The relationship between protein domains and homopeptides in the Plasmodium falciparum proteome

Wang Yue
Yang Hsin Jou
Harrison Paul M. paul.harrison@mcgill.ca
Department of Biology, McGill University , Montreal, QC , Canada
Gillespie Joseph
Electronic publication date: 2020 Oct 2
Publication date: 2020
Volume: 8
Electronic Location ID: e9940
Received 2020 Apr 30; Accepted 2020 Aug 24
Copyright: © 2020 Wang et al.
Copyright year: 2020
Copyright holder: Wang et al.
License: This is an open access article distributed under the terms of the Creative Commons Attribution License, which permits unrestricted use, distribution, reproduction and adaptation in any medium and for any purpose provided that it is properly attributed. For attribution, the original author(s), title, publication source (PeerJ) and either DOI or URL of the article must be cited.
License URL: https://creativecommons.org/licenses/by/4.0/

Keywords: Homopeptide, Low-complexity, Plasmodium, Intrinsic disorder, Asparagine, Protein domains

Funding: Natural Sciences and Engineering Research Council of Canada This work was supported by the Natural Sciences and Engineering Research Council of Canada. The funders had no role in study design, data collection and analysis, decision to publish, or preparation of the manuscript.

==============================
The proteome of the malaria parasite Plasmodium falciparum is notable for the pervasive occurrence of homopeptides or low-complexity regions (i.e., regions that are made from a small subset of amino-acid residue types). The most prevalent of these are made from residues encoded by adenine/thymidine (AT)-rich codons, in particular asparagine. We examined homopeptide occurrences within protein domains in P. falciparum. Homopeptide enrichments occur for hydrophobic (e.g., valine), or small residues (alanine or glycine) in short spans (<5 residues), but these enrichments disappear for longer lengths. We observe that short asparagine homopeptides (<10 residues long) have a dramatic relative depletion inside protein domains, indicating some selective constraint to keep them from forming. We surmise that this is possibly linked to co-translational protein folding, although there are specific protein domains that are enriched in longer asparagine homopeptides (≥10 residues) indicating a functional linkage for specific poly-asparagine tracts. Top gene ontology functional category enrichments for homopeptides associated with diverse protein domains include “vesicle-mediated transport”, and “DNA-directed 5′-3′ RNA polymerase activity”, with various categories linked to “binding” evidencing significant homopeptide depletions. Also, in general homopeptides are substantially enriched in the parts of protein domains that are near/in IDRs. The implications of these findings are discussed.

Introduction

Plasmodium falciparum (Pf) is a single-celled protozoan that causes malaria in humans. Malaria causes hundreds of thousands of deaths every year, with ~405,000 in 2018 (Global Malaria Programme, 2019). Treatment for malaria is confounded by its ability to adapt quickly to drugs and to the human immune system; its antigenic diversity is a major problem for vaccine development (Ferreira, Da Silva Nunes & Wunderlich, 2004; Ferreira et al., 2003; Freitas-Junior et al., 2000). The complete genome sequence of Pf contains >5,000 protein-coding genes (Gardner et al., 2002). Early analysis indicated that low-complexity regions (LCRs) (i.e., regions that consist mostly of a small subset of amino-acid types) or homopeptides (runs of single amino acids) are a prominent feature of the encoded proteins, with more than half of proteins being low-complexity over most of their sequences (Pizzi & Frontali, 2001). Asparagine-rich regions are the most abundant (An & Harrison, 2016; Pizzi & Frontali, 2001). The LCRs have been postulated to have a function primarily at the nucleotide level (Xue & Forsdyke, 2003). Their abundance depends largely on genomic A+T or G+C content (DePristo, Zilversmit & Hartl, 2006; Xue & Forsdyke, 2003), and they also acquire further low-complexity insertions and deletions according to a power-law rule: that is, longer LCRs acquire longer insertions/deletions (DePristo, Zilversmit & Hartl, 2006). Pf LCRs can be classified into three distinct types, including a high G+C type that is linked to recombination hotspots (Zilversmit et al., 2010). There is a pattern of enrichment of long intergenic poly(AT) tracts in Plasmodium species, some of which are immediately adjacent to genes and run into them (Russell et al., 2014). Although asparagine is preferred in LCRs of Pf, a different residue type with AT-rich codons (lysine) is more prominent in the CVK group, which is a set of four primate-infecting plasmodia (Chaudhry et al., 2018). As well as being sites of polymorphic variation themselves (Chaudhry et al., 2018), Pf LCRs are linked to increased single-nucleotide polymorphism in their vicinity (Haerty & Golding, 2011).

Although homopeptides are sites of such polymorphic variation, earlier work showed that some homopeptides are deeply conserved across orthologs from bacteria and eukaryotes, suggesting ancient origin and functional essentiality (Faux et al., 2005). In general, homopeptides are more conserved in bacteria, than in archaea and eukaryotes, and there is a correlation between repeat length differences and species divergence (Uthayakumar et al., 2012). Homopeptides increase the functional versatility of proteins, and facilitate spatial organization of proteins in a repeat-dependent way (Chavali et al., 2017). They are also significantly linked to many human diseases (Lobanov et al., 2016).

Low-complexity regions rich in hydrophilic residues are significantly associated with protein intrinsic disorder (Delucchi et al., 2020; Romero et al., 2001). Previous surveys have shown that 10–40% of Pf LCR residues are predicted as intrinsically disordered in tracts ≥40 residues long, and that >60% of sequences have such a tract (Feng et al., 2006; Mohan et al., 2008). Such annotated disordered regions in Pf are significantly depleted of predicted MHC-binding peptides, which has implications for vaccine development, since many vaccine target proteins are intrinsically disordered (Guy et al., 2015).

Low-complexity regions rich in asparagine (and/or glutamine) are common in domains that form prions (i.e., self-propagating amyloid particles) (Harbi & Harrison, 2014a; Harbi et al., 2012; Harrison, 2017; Su & Harrison, 2019). In budding yeast (Saccharomyces cerevisiae), propagation of these particles can be sustained during budding, mating and laboratory protocols (Harbi & Harrison, 2014b). Predicted prions have been detected in all the domains of life (Espinosa Angarica, Ventura & Sancho, 2013), including thousands in viruses and phages (Tetz & Tetz, 2017; 2018), and tens of thousands in bacteria (Harrison, 2019). Pf has prion-like domains (that arise in asparagine-rich LCRs) in 10–24% of its proteins (Singh et al., 2004; An & Harrison, 2016; Pallares et al., 2018). Just like Pf, there are Saccharomycetes fungi that have high proportions of prion-like proteins with poly-asparagine in them (An, Fitzpatrick & Harrison, 2016). There is some evidence that asparagine-rich LCRs act as “tRNA sponges” that slow down the translation rate of proteins to aid in co-translational folding (Filisetti et al., 2013; Frugier et al., 2010). This is a type of parallel function at the DNA/RNA level for which there is increasing evidence for intrinsically disordered regions (IDRs) in proteins (Pancsa & Tompa, 2016).

Here, we investigate the relationship between homopeptides and both defined protein domains and intrinsic disorder in Pf proteins. We observe significant depletions in homopeptides for specific types of amino acid, in particular asparagine and aspartate. Homopeptides are substantially enriched in the parts of protein domains that are near or in IDRs.

Methods

Source data

The UniProt (Boeckmann et al., 2003) reference proteome for Plasmodium falciparum strain 3D7 was downloaded from www.uniprot.org in January 2019. Protein domain annotations for Pf were taken from the Pfam database (El-Gebali et al., 2019). For comparative analysis, three further reference proteomes were downloaded from the same source for the following: another P. falciparum strain (FCH/4), P. yoelii (strain 17XNL), and P. vivax (strain Salvador I).

Annotation

Homopeptides were defined as repetitions of one amino-acid type with a minimum length of three residues (Fig. 1).

Figure 1 Schematic of the analysis.

Homopeptides were defined as ≥3 consecutive amino acids of the same type in a sequence. Protein domains near intrinsically disordered regions (IDRs) were determined using a 10-residues buffer. Also, if the IDR is within a protein domain or otherwise overlaps it, a 10-residue buffer is considered on either side of the IDR as shown.

Proteins were annotated for intrinsic disorder using the DISOPRED3 and IUPRED2a programs (Dosztanyi et al., 2005; Huntley et al., 2015; Ward et al., 2004). IUPRED2a operates on inputted single sequences, and predicts intrinsic disorder by estimating inter-residue interaction energies (Erdos & Dosztanyi, 2020). It was the best performing single-sequence method for intrinsic disorder annotation, with an area-under-curve (AUC) value of 0.83 for the ROC curve in a recent assessment (Nielsen & Mulder, 2019); DISOPRED3 also had a value of AUC = 0.83 in this assessment, and was one of the best performing methods that use evolutionary information as input. Only regions of predicted disorder ≥30 residues long were considered. A 30-residue length cut-off was used since this is a common threshold or boundary value used in characterizing intrinsically-disordered regions, or in training algorithms for prediction of intrinsic disorder (Atkins et al., 2015). Thus, we used the default “long” parameter choice for the IUPRED2a program. Also, the DISOPRED3 program was run with a 2% expected false positive rate for the algorithm training set, which is in is the author’s recommended parameter range (Jones & Cozzetto, 2015). The results from either intrinsic disorder annotation program were considered separately.

Enrichments & depletions

Because we wish to examine the effects of protein domain structure and intrinsic disorder on the occurrence of homopeptides in Pf, we checked whether there is any deviation from random placement for homopeptides within protein domains and annotated intrinsic disorder. These can be either enrichments relative to background populations or depletions. The background populations were either the whole proteome or the set of protein domain annotations as described below. Enrichments and depletions for homopeptides in protein domains were calculated as depicted (Fig. 1). These were determined for individual amino-acid types in homopeptides using Eq. (1) below for hypergeometric probability, for sampling with replacement: (1) P(k)=(Kk)(N−Kn−k)(Nn)

with the sample counts given by:

k = number of residues in homopeptides in domains of one amino-acid type

n = number of residues in homopeptides in domains

and the background counts given by the quantities:

K = number of residues in homopeptides in the proteome of one amino-acid type

N = number of residues in homopeptides in the proteome

Enrichments/depletions for the amount of homopeptides in specific protein domain types were also calculated with the sample counts given by:

k = number of residues in homopeptides in one domain type

n = number of residues in homopeptides in all domain types

and the background counts given by the quantities:

K = number of residues in one domain type

N = number of residues in all domain-types

Enrichments and depletions for protein domains overlapping or near annotated IDRs were also calculated with the sample counts given by:

k = number of residues in one domain type which near or inside disordered regions

n = number of residues in all domain types which near or inside disordered regions

and the background counts given by the quantities:

K = number of residues in one domain type

N = number of residues in all domain-types

Proximity to IDRs was determined using a 10-residue buffer at either end of the annotated IDRs (Fig. 1).

Gene ontology (GO) category enrichments were also analyzed for specific types of homopeptide enrichment (Huntley et al., 2015). These were calculated by mapping the protein domains onto GO categories, and re-totalling the numbers of residues per GO category rather than per domain.

All enrichments/depletions were calculated using hypergeometric probability with appropriate Bonferroni corrections for multiple hypothesis testing. For example, the Bonferroni correction for enrichments/depletions of homopeptides of individual amino acids in protein domains was P = 0.05/20 = 0.0025, since there are 20 different amino acids being sampled from the same background population.

Propensity

A propensity for homopeptides of a specific amino-acid type to occur in protein domains (Pdom) was calculated as: (2) Pdom=log10[(k/n)/(K/N)]

The values of k, n, K and N are as listed above just below Eq. (1). This was calculated for the homopeptide threshold ≥3 residues.

Results

Enrichments and depletions of homopeptides in protein domains

Homopeptides are abundant and pervasive in the Pf proteome, yet it is not clear from a structural perspective which homopeptides are more tolerated in protein domains. We analyzed the preferences of homopeptides of each specific amino-acid type for insertion into protein domains, using three different length thresholds for homopeptides (Table 1). The statistical enrichments/depletions are listed, as well as the fraction of the homopeptide populations for each amino acid, and the propensity (Pdom) of homopeptides of each amino acid for protein domains, calculated as described in “Methods”. For the minimum homopeptide threshold of ≥3 residues length, the most enriched amino acids include the major aliphatic hydrophobic residues valine, isoleucine and leucine, which is to be expected because of the extensive hydrophobic cores of protein domains. Also, the small residues alanine and glycine exhibit highly significant enrichments. For these amino-acid types, the enrichments are only for short homopeptides (of size 3 or 4 residues), since the enrichments disappear for longer homopeptide thresholds (Table 1). Positively-charged homopeptides and other hydrophilic homopeptides are also generally enriched (lysine, arginine, serine, and threonine), while negatively-charged homopeptides are significantly depleted or show no preferences. Lysine homopeptides are the second most abundant in the proteome and are made from AT-rich codons; their relationships with specific protein domains are discussed below. Some amino acids show an enrichment, with a comparable propensity for structural domains (Pdom) as for other amino acids, but these are not significant. Most strikingly though, short asparagine homopeptides are highly significantly depleted within protein domains.

Table 1 Homopeptides enriched/depleted amino-acid types in protein domains sorted by P-values.

Amino Acid (one letter code)	Homopeptide amount (≥3) in pfam domain (n = 19,052)	Fraction in domains	Pdom***	Homopeptide amount(≥3) in proteome (N = 209,236)	P-value*	Homopeptide amount (≥5) in pfam domain (n = 2,121)	Homopeptide amount (≥5) in proteome (N = 61,333)	P-value	Homopeptide amount (≥10) in pfam domain (n = 349)	Homopeptide amount (≥10) in proteome (N = 17,724)	P-value	
A	723	0.76	+0.92	954	0	0	0	1	0	0	1	
L	1,676	0.20	+0.34	8,250	2.44e−222	0	261	0.00010	0	0	1	
I	1,527	0.21	+0.36	7,315	4.69e−214	0	0	1	0	0	1	
G	579	0.38	+0.62	1,527	1.30e−206	0	125	0.01223	0	0	1	
V	378	0.34	+0.57	1,108	4.13e−118	0	0	1	0	0	1	
R	454	0.28	+0.49	1,606	1.72e−108	0	121	0.01408	0	0	1	
T	417	0.18	+0.30	2,341	2.67e−40	15	318	0.05316	0	0	1	
K	5,036	0.10	+0.04	48,223	1.81e−31	509	1,,0312	1.95e−18	0	0	1	
S	1,059	0.11	+0.08	9,511	9.57e−13	75	1,167	1.51e−07	0	146	0.05417	
P	131	0.17	+0.27	774	2.45e−12	0	129	0.01062	0	0	1	
F	481	0.12	+0.12	3,913	4.00e−12	0	144	0.00626	0	0	1	
Y	393	0.11	+0.08	3,582	1.54e−05	0	190	0.00123	0	0	1	
C	27	0.18	+0.30	148	0.00022	0	5	0.83864	0	0	1	
Q	121	0.11	+0.08	1098	0.00384	15	256	0.01668	0	107	0.11832	
E	1170	0.09	–0.01	12572	0.00914	65	1907	0.05070	0	380	0.00048	
W	3	0.33	+0.56	9	0.03576	0	0	1	0	0	1	
M	22	0.12	+0.12	189	0.04567	0	0	1	0	0	1	
N**	3,941	0.04	−0.36	92,722	0	1,363	43,632	5.11e−13	349	16,627	1.65e−10	
D	881	0.07	−0.11	12,636	7.83e−20	79	2642	0.01802	0	464	8.68e−05	
H	33	0.04	−0.36	758	2.31e−07	0	124	0.01267	0	0	1	
Notes:

* P-value threshold = 0.0025 (with a Bonferroni correction accounting for tests on the twenty amino acids). P-values of 0.0 are infinitesimally small beyond the precision of the computation.

** Significant enrichments or depletions are in bold. Underlined ones are homopeptide-depleted amino acids.

*** Pdom is the propensity of homopeptides of a specific amino-acid type to occur in protein domains. It is calculated as described in “Methods”.

Histograms of homopeptide length also indicate that within protein domains, homopeptides generally lack the longer homopeptide lengths (≥10 residues) that make up the majority of homopeptides outside of protein domains (Fig. 2).

Figure 2 Distribution of homopeptide length inside and outside of protein domains.

(A) The distribution of homopeptide length for all residues both inside and outside protein domains. The natural log of the total number of homopeptides for a given length is used. (B–D) are the same distributions but for K-, N-, and D-homopeptides, respectively.

For the longest homopeptide threshold (≥10 residues length), the number of amino-acid types which are comparatively tolerated in protein domains dramatically decreases to one (asparagine; Table 1). Keeping in mind that the enrichment calculations are effectively based on the comparison of different amino-acid types, there should always be at least one enriched amino-acid type unless there are completely no homopeptides at all at a certain threshold. Poly-asparagine homopeptides are depleted in domains until the threshold is extended to 10, which leaves it as the only one existing in domains. The enrichment observed for longer polyasparagine tracts (≥10 residues, Table 1) arises from a small number of specific protein domains that may have a functional linkage for these polyasparagine tracts (Table S1), for example, for specific protein interactions.

The enrichment/depletion results for individual amino-acid types are little affected by the boundary definition of protein domains (i.e., chopping off 3, 5 or 7 residues from the ends of the domains, Table S3), with just some enrichments for glutamine and glutamate becoming significant for these shortened domains. This indicates that these homopeptides significantly occur near the ends of protein domains.

Since lysine and arginine homopeptides are in general significantly enriched in protein domains, and asparagine and aspartate significantly depleted, we examined which individual protein domains are linked to these trends (Table S1). Despite the substantial general depletion of asparagine homopeptides within protein domains, there are 87 individual domains with significant enrichment of asparagine homopeptides, including the low-copy-number Sin-like region and the SacI homology domain; these also stand out when we restrict the analysis to polyasparagine tracts ≥10 residues long (Table S1). The most prominent lysine homopeptide enrichments are for Rifin and PfEMP DBL domains. Specific domains are also linked to hydrophobic or small-residue homopeptides, such as glycine homopeptides arising for ribosomal proteins (Table S1).

Gene ontology enrichments

We examined the enrichments and depletions of GO functional categories associated with homopeptides in protein domains (Table S2). Some top enrichments of GO functional categories for homopeptides in protein domains include: GO:0003899 (DNA-directed 5′-3′ RNA polymerase activity), an enrichment caused by seven different protein domains, and GO:0042578 (phosphoric ester hydrolase activity), which is unique to the SacI homology domain (which is involved in clathrin-mediated endocytosis; three copies in Pf) (Table S2). Inspection of other GO category enrichments indicate that they are also caused by diverse protein domains, for example, GO:0016192 (vesicle-mediated transport) which is linked to homopeptide enrichments in 11 different protein domains, pointing to specific functional significance for homopeptides in the interaction of these proteins. Nonetheless, in general “protein binding” is significantly depleted in the list (Table S2), as are the other high-level “RNA-binding” and “GTP-binding” terms.

The relationship between specific protein domain homopeptides and intrinsic disorder

Intrinsically disordered regions tend to have homopeptides and low-complexity sequences in them (Romero et al., 2001). We surmised that the relationship of different protein domains with homopeptides might be caused by their proximity to or overlap with IDRs of proteins. In general, homopeptides are enriched in the parts of protein domains that are near or in IDRs (Table 2; results for either the DISOPRED3 or IUPred2A program are shown). Also, there is only one individual protein domain type that is significantly depleted in homopeptides near/in IDRs, with the remainder of significant deviations being enrichments (Table 2).

Table 2 Enrichment of homopeptides within protein domains that are near or overlapping IDRs.

Intrinsic disorder annotator	Total number of domain residues in/near IDRs	Homopeptide residues in/near IDRs	Total number of domain residues	Total number of homopeptides in domains	P-value*	P-value**	P-value***	
IUPred2A	38,940	2,845	808,565	19,052	0.0	49	1	
DisoPred3	15,928	1,381	808,565	19,052	0.0	29	1	
Notes:

* These P-value results are not affected by chopping off 3, 5 or 7 residues from the ends of the protein domains, as for Table S3.

** Total number of individual protein domains that have enrichment of homopeptides within their parts that are near or overlapping IDRs.

*** Total number of individual protein domains that have depletion of homopeptides within their parts that are near or overlapping IDRs.

Comparison of trends in other plasmodia

The trends observed for Pf strain 3D7 were validated by analysis of another Pf strain (FCH/4) that was picked from the UniProt reference proteome list (Boeckmann et al., 2003) (Table S4). There is just one small change with enrichments of glutamate homopeptides in protein domains becoming significant (Table S4). Comparisons were also made with proteomes of P. yoelii, a malaria parasite of rodents, and P. vivax, a member of the CVK group of primate-infecting plasmodia (Chaudhry et al., 2018). P. yoelii has an overall approximately even predominance of N and K homopeptides, and P. vivax has predominance of K homopeptides rather than of N homopeptides (Table S4). The depletion of N homopeptides in protein domains is maintained in P. yoelii, but there is no significant depletion/enrichment in P. vivax. K homopeptides also become significantly depleted within protein domains in P. yoelii, despite their similar overall levels to Pf (29% in P. yoelii vs 23% in Pf). The results for homopeptide enrichments in parts of protein domains overlapping IDRs (Table 2) also remain highly significant for these three other Plasmodia proteomes (P-values ~ 0.0).

Discussion

Homopeptide trends

Homopeptide enrichments within protein domains, such as for hydrophobic (L, I or V) or small (A or G) residues, disappear at longer lengths (≥5 residues). This indicates a limit to their toleration within protein domain cores, for example, because they are not so easily accommodated in regular secondary structures.

We observed a substantial significant relative depletion of short asparagine runs (<10 residues long) in protein domains. Plasmodia have acquired great amounts of N homopeptide tracts during evolution, but statistically these have not been appearing or “landing” within domains. The lack of short intra-domain asparagine runs may be because they interfere with protein folding in some way. For example, they may slow down co-translational protein folding due to a lack of asparaginyl-tRNAs, since levels of asparaginyl-tRNAs in Pf are normal despite the high amounts of asparagine in their coding sequences (Filisetti et al., 2013; Frugier et al., 2010). Thus asparagine homopeptides may be “tRNA sponges” that soak up tRNAs and slow down translation and co-translational folding (Filisetti et al., 2013; Frugier et al., 2010). It is possible that homopeptides, and in particular poly-asparagine homopeptides may make protein domains more prone to misfolding. Although generally slower translation is thought to aid in correct co-translational folding (Waudby, Dobson & Christodoulou, 2019), sometimes faster translation is more desirable through segments that are prone to misfolding (O’Brien, Vendruscolo & Dobson, 2014), or for translational efficiency at buried residue sites or sites that are vulnerable to structurally disruptive mutations (Wang et al., 2015; Zhou, Weems & Wilke, 2009). However, experiments with Pf chaperone Hsp110c, have shown that Pf has cellular mechanisms that are designed to prevent aggregation linked to asparagine tracts (Muralidharan & Goldberg, 2013). A few specific protein domains have enrichment of long polyasparagine tracts. Such tracts may have a specific functional role in these proteins, perhaps for protein or nucleic-acid interaction. Another possibility might be that correct folding of these specific domains is not affected by slow rates of translation, thus N homopeptides can arise in them because of their general abundance in the proteome. Interestingly, the significant depletion of N homopeptides is maintained in the rodent malaria pathogen P. yoelii, but there is an absence of significant depletion/enrichment in the more distantly related P. vivax malaria pathogen from the CVK group. This may indicate that the N homopeptides in the P. vivax protein domains are reduced mainly to those that have functional roles.

Gene ontology enrichments

In the GO enrichments/depletion analysis we see a general trend for depletion of functional categories associated with “binding” (protein binding; RNA binding; GTP binding; ion binding). This suggests that homopeptides may be selected against in structured interaction interfaces, perhaps since they introduce a lack of interaction specificity, or increase the likelihood of off-target binding. Also, the GO results indicate that homopeptide occurrences may be useful information for the discrimination of protein function from the analysis of sequences (Huntley et al., 2014, 2015; Jiang et al., 2016; Le et al., 2019; Le, Yapp & Yeh, 2019; Mutowo-Meullenet et al., 2013).

Intrinsic disorder

We surmise that homopeptides in protein domains are enriched in the parts of the domains near or in IDRs because IDRs generally have more tolerance for insertions/deletions, and are the main determinants of changes in protein length over evolution (Light et al., 2013). Also, our results indicate that the parts of protein domains that can become intrinsically disordered are enriched in homopeptides relative to other domain parts. This may be an important part of encrypting their ability to transition structurally (Narasumani & Harrison, 2015).

Conclusions

The most pervasive homopeptide in Plasmodium falciparium, poly-asparagine, is substantially depleted within protein domains, whereas other homopeptides that we might expect in the hydrophobic core of domains, such as poly-leucine or -valine or -isoleucine, and other generally abundant homopeptides (such as lysine) are enriched at the shorter homopeptide lengths studied. We hypothesize that generally poly-asparagine formation is repressed inside protein domains because its occurrence may slow co-translational folding (Filisetti et al., 2013; Frugier et al., 2010), which might be problematic within a protein domain that has only partially been translated (Scenarios in which both transient fast and slow folding may be problematic for co-translational protein folding are possible (O’Brien, Vendruscolo & Dobson, 2014)). Further experimental work is needed to investigate these hypotheses. In general, homopeptides are depleted for functional categories associated with diverse types of binding, indicating that they may interfere with specificity in structured interfaces. However, the parts of protein domains that can become intrinsically disordered have homopeptide enrichment relative to other parts of domains. Since some domains fold upon binding to other proteins, and the parts of protein domains that overlap intrinsic disorder have such homopeptide enrichment, these results suggest that protein homopeptides may be useful in effecting such structural transitions.

Supplemental Information

Supplemental Information 1 Protein domains that are enriched/depleted in specific homopeptides (≥3 length: A, L, I, G, K, R, N, D and ≥10: N).

Click here for additional data file.

Supplemental Information 2 Full list of enrichments and depletions of homopeptide enrichments/depletions for Gene Ontology categories.

Click here for additional data file.

Supplemental Information 3 Homopeptide enriched/depleted amino-acid types in protein domains, with various degrees of protein domain shortening.

The format of this table is modeled on Table 1.

Click here for additional data file.

Supplemental Information 4 Homopeptide enriched/depleted amino-acid types in protein domains in other Plasmodium proteomes.

The format of this table is modeled on Table 1.

Click here for additional data file.

Additional Information and Declarations

Competing Interests

Author Contributions

Data Availability

The authors declare that they have no competing interests.

Yue Wang analyzed the data, prepared figures and/or tables, and approved the final draft.

Hsin Jou Yang analyzed the data, prepared figures and/or tables, and approved the final draft.

Paul M. Harrison analyzed the data, prepared figures and/or tables, authored or reviewed drafts of the paper, and approved the final draft.

The following information was supplied regarding data availability:

The data are available in the Supplemental Files.

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
