# Peer review of "The relationship between protein domains and homopeptides in the Plasmodium falciparum proteome"

_PeerJ, doi:10.7717/peerj.9940_

## Round 0.1 · original submission · Major Revisions

Dear Dr. Wang and colleagues:

Thanks for submitting your manuscript to PeerJ. I have now received three independent reviews of your work, and as you will see, the reviewers raised some concerns about the research. Despite this, these reviewers are optimistic about your work and the potential impact it will have on research studying protein domains and homopeptides in the Plasmodium falciparum proteome. Thus, I encourage you to revise your manuscript, accordingly, taking into account all of the concerns raised by both reviewers.

While the concerns of the reviewers are relatively minor, this is a major revision to ensure that the original reviewers have a chance to evaluate your responses to their concerns. There are many suggestions, which I am sure will greatly improve your manuscript once addressed.

I look forward to seeing your revision, and thanks again for submitting your work to PeerJ.

Good luck with your revision,

-joe

Reviewer 1 ·

Basic reporting

The study is quite thoughtfully designed and the statistical analyses complement the results very well. The fact that homopeptides are more prominently present in the proteome but outside domains rather than within is a substantial find and is well supported with data. The fact about most homopeptides being enriched in shorter tracts while a long polyN tract is tolerated is most definitely an important result.
Language used is clear and well-articulated. Figures could be better analysed though and the study could have gone into more depth.

Experimental design

Research question and methodology well defined. At the same time, should have explained the statistical analyses rationale as well, but that is alright. If the results showed the percentage of relative depletions between the presence of homopeptides within domains and those outside – it would have been better. The fact that the total numbers were not normalised as to see what percent of the homopeptides of a particular nature of a particular length were within which segment – outside or within the domains needs to be revised. Those numbers would make the differences more stark and make the claims more convincing. The statistical analyses performed are of a good standard and there are no issues there.

Validity of the findings

The finding about depleted/enriched homopeptides at various lengths for differing amino acids is indeed valid although, a clear value of depletion/enrichment ratios would have made the finding stronger. The hypothesis about poly-N tracts being repressed within domains in comparison to outside it and throughout the genome may be considered a decent claim until proven otherwise. This affecting slowing down of co-translational folding is a legit biochemical possibility but it needs more evidence to be validated. The finding about IDRs being more enriched in homopeptides due to their higher tolerance is a possibility but the authors show no data about the relative positions of the homopeptides they actually found within the domains – alignments or schematics missing so…this is a slippery slope, although it has been shown by Russel et al, 2014 that homopeptides actually demarcate the margins of protein domains and sometimes run into them…this does not validate claims about specificity and transitional changes for differing activities. Finally, reframing the last line of conclusions in a more subtle tone could better convey the message.

Reviewer 2 ·

Basic reporting

Here Wang et al. investigated the association of homopeptide in protein domains within human malaria proteome. The manuscript is written well and covered all aspects of the area that need to be reviewed before publication. I do not have any additional comments and overall, I recommend this article for publication.

Experimental design

no comment

Validity of the findings

no comment

Additional comments

Line 41. Please quote the correct number of deaths annually caused by malaria from WHO reference.

Reviewer 3 ·

Basic reporting

- It is important to add more literature references on the performance of previously similar works.No

Experimental design

- In equation 1, what is "X"?

Validity of the findings

- It is important to validate the results on an external dataset.
- Statistical tests have not been explained well. There is a need for adding more description in this part.
- The authors used two published methods for annotating intrinsic disorder. What are the correctness and effective of these methods? It is an important step to make sure that the authors label data correctly.
- If the authors used two methods (DISOPRED3 and IUPRED2a), did they perform experiments separately or merge together?
- Why did the authors not set any parameters of these algorithms?
- UniProt or GO has been used in previous works in biomedical such as PMID: 31921391 and PMID: 31277574. Therefore, the authors should add more references in this description.
- In Table 1, significant values are highlight. Why did the amino acid "E" have low p-value (0.00048), but "E" is not considered as significant?

Additional comments

- Equations need to be assigned their numbers.
- Language writing needs to be improved. Also, there are some typos in the manuscript.

---

## Round 0.2 · accepted · Accept

Dear Dr. Wang and colleagues:

Thanks for revising your manuscript based on the concerns raised by the reviewers. I now believe that your manuscript is suitable for publication. Congratulations! I look forward to seeing this work in print, and I anticipate it being an important resource for groups studying protein domains and homopeptides in the Plasmodium falciparum proteome. Thanks again for choosing PeerJ to publish such important work.

Best,

-joe

Reviewer 2 ·

Basic reporting

no comment

Experimental design

no comment

Validity of the findings

no comment

Additional comments

I have read the revised version of the manuscript and the point-by-point rebuttal provided by the authors and I feel that most of the comments and suggestions have been well transferred in the current version of the manuscript. I do not have any additional comments and would favor publication of this study.

Reviewer 3 ·

Basic reporting

My previous comments have been addressed well.

Experimental design

My previous comments have been addressed well.

Validity of the findings

My previous comments have been addressed well.

Additional comments

My previous comments have been addressed well.